# Time2Vec: Learning a Vector Representation of Time

## Abstract

Time is an important feature in many applications involving events that occur synchronously and/or asynchronously. To effectively consume time information, recent studies have focused on designing new architectures. In this paper, we take an orthogonal but complementary approach by providing a model-agnostic vector representation for time, called *Time2Vec*, that can be easily imported into many existing and future architectures and improve their performances. We show on a range of models and problems that replacing the notion of time with its Time2Vec representation improves the performance of the final model.

## 1 Introduction

In building machine learning models, "time" is often an important feature. Examples include predicting daily sales for a company based on the date (and other available features), predicting the time for a patient's next health event based on their medical history, and predicting the song a person is interested in listening to based on their listening history. The input for problems involving time can be considered as a sequence where, rather than being identically and independently distributed (*iid*), there exists a dependence across time (and/or space) among the data points. The sequence can be either synchronous, *i.e.* sampled at regular intervals, or asynchronous, *i.e.* sampled at different points in time. In both cases, time may be an important feature. For predicting daily sales, for instance, it may be useful to know if it is a holiday or not. For predicting the time for a patient's next encounter, it is important to know the (asynchronous) times of their previous visits.

Recurrent neural networks (RNNs) do not typically treat time itself as a feature, typically assuming that inputs are synchronous. When time is known to be a relevant feature, it is often fed in as yet another input dimension (Choi et al., 2016; Du et al., 2016; Li et al., 2018b). In practice, RNNs often fail at effectively making use of time as a feature. To help the RNN make better use of time, several researchers design hand-crafted features of time that suit their specific problem and feed those features into the RNN (Choi et al., 2016; Baytas et al., 2017; Kwon et al., 2019). Hand-crafting features, however, can be expensive and requires domain expertise about the problem.

Many recent studies aim at obviating the need for hand-crafting features by proposing general-purpose—as opposed to problem specific—architectures that better handle time (Neil et al., 2016; Zhu et al., 2017; Mei & Eisner, 2017; Hu & Qi, 2017; Upadhyay et al., 2018; Li et al., 2018a). We follow an orthogonal but complementary approach to these recent studies by developing a general-purpose model-agnostic representation for time that can be potentially used in any architecture. In particular, we develop a learnable vector representation (or embedding) for time as a vector representation can be easily combined with many models or architectures. We call this vector representation *Time2Vec*. To validate the effectiveness of Time2Vec, we conduct experiments on several (synthesized and real-world) datasets and integrate it with several architectures. Our main result is to show that on a range of problems and architectures that consume time, using Time2Vec instead of the time itself offers a boost in performance.

## 2 Related Work

There is a long history of algorithms for predictive modeling in time series analysis. They include auto-regressive techniques (Akaike, 1969) that predict future measurements in a sequence based on a

window of past measurements. Since it is not always clear how long the window of past measurements should be, hidden Markov models (Rabiner & Juang, 1986), dynamic Bayesian networks (Murphy & Russell, 2002), and dynamic conditional random fields (Sutton et al., 2007) use hidden states as a finite memory that can remember information arbitrarily far in the past. These models can be seen as special cases of recurrent neural networks (Hochreiter & Schmidhuber, 1997). They typically assume that inputs are synchronous, *i.e.* arrive at regular time intervals, and that the underlying process is stationary with respect to time. It is possible to aggregate asynchronous events into time-bins and to use synchronous models over the bins (Lipton et al., 2016; Anumula et al., 2018). Asynchronous events can also be directly modeled with point processes (*e.g.*, Poisson, Cox, and Hawkes point processes) (Daley & Vere-Jones, 2007; Laub et al., 2015; Xiao et al., 2017; Li et al., 2018a; Xiao et al., 2018) and continuous time normalizing flows (Chen et al., 2018). Alternatively, one can also interpolate or make predictions at arbitrary time stamps with Gaussian processes (Rasmussen, 2004) or support vector regression (Drucker et al., 1997).

Our goal is not to propose a new model for time series analysis, but instead to propose a representation of time in the form of a vector embedding that can be used by many models. Vector embedding has been previously successfully used for other domains such as text (Mikolov et al., 2013; Pennington et al., 2014), (knowledge) graphs (Grover & Leskovec, 2016; Nickel et al., 2016; Kazemi & Poole, 2018), and positions (Vaswani et al., 2017; Gehring et al., 2017). Our approach is related to time decomposition techniques that encode a temporal signal into a set of frequencies (Cohen, 1995). However, instead of using a fixed set of frequencies as in Fourier transforms (Bracewell & Bracewell, 1986), we allow the frequencies to be learned. We take inspiration from the neural decomposition of Godfrey & Gashler (2018) (and similarly (Gashler & Ashmore, 2016)). For time-series analysis, Godfrey & Gashler (2018) decompose a 1D signal of time into several sine functions and a linear function to extrapolate (or interpolate) the given signal. We follow a similar intuition but instead of decomposing a 1D signal of time into its components, we transform the time itself and feed its transformation into the model that is to consume the time information. Our approach corresponds to the technique of Godfrey & Gashler (2018) when applied to regression tasks in 1D signals, but it is more general since we learn a representation that can be shared across many signals and can be fed to many models for tasks beyond regression.

While there is a body of literature on designing neural networks with sine activations (Lapedes & Farber, 1987; Sopena et al., 1999; Wong et al., 2002; Mingo et al., 2004; Liu et al., 2016), our work uses sine only for transforming time; the rest of the network uses other activations. There is also a set of techniques that consider time as yet another feature and concatenate time (or some hand designed features of time such as log and/or inverse of delta time) with the input (Choi et al., 2016; Li et al., 2017; Du et al., 2016; Baytas et al., 2017; Kwon et al., 2019; Trivedi et al., 2017; Kumar et al., 2018; Ma et al., 2018). Kazemi et al. (2019) survey several such approaches for dynamic (knowledge) graphs. These models can directly benefit from our proposed vector embedding, Time2Vec, by concatenating Time2Vec with the input instead of their time features. Other works (Neil et al., 2016; Zhu et al., 2017; Mei & Eisner, 2017; Hu & Qi, 2017; Upadhyay et al., 2018; Li et al., 2018a) propose new neural architectures that take into account time (or some features of time). We show how Time2Vec can be used in one of these architectures to better exploit temporal information; it can be potentially used in other architectures as well.

## 3   BACKGROUND & NOTATION

We use lower-case letters to denote scalars, bold lower-case letters to denote vectors, and bold upper-case letters to denote matrices. We represent the $i^{th}$ element of the vector $r$ as $r[i]$. For two vectors $r$ and $s$, we use $[r; s]$ to represent their concatenation and $r \odot s$ to represent element-wise (Hadamard) multiplication of the two vectors. Throughout the paper, we use $\tau$ to represent a scalar notion of time (*e.g.*, absolute time or time from the last event) and $\boldsymbol{\tau}$ for a vector of time features.

Long Short Term Memory (LSTM) (Hochreiter & Schmidhuber, 1997) is considered one of the most successful RNN architectures for sequence modeling. A formulation of the original LSTM model and a variant of it based on peepholes (Gers & Schmidhuber, 2000) is presented in Appendix C. When time is a relevant feature, the easiest way to handle time is to consider it as just another feature (or extract some engineered features from it), concatenate the time features with the input, and use the standard LSTM model (or some other sequence model) (Choi et al., 2016; Du et al., 2016; Li et al.,

2018b). In this paper, we call this model *LSTM+T*. Another way of handling time is by changing the formulation of the standard LSTM. Zhu et al. (2017) developed one such formulation, named *TimeLSTM*, by adding time gates to the architecture of the LSTM with peepholes. They proposed three architectures namely *TLSTM1, TLSTM2, TLSTM3*. A description of *TLSTM1* and *TLSTM3* can be found in Appendix C (we skipped TLSTM2 as it is quite similar to TLSTM3).

## 4 TIME2VEC

A common approach to deal with time in different applications is to apply some hand-crafted function(s) $f_1, \ldots, f_m$ to $\tau$ ($\tau$ can be absolute time, time from last event, etc.), concatenate the outputs $f_1(\tau), \ldots, f_m(\tau)$ with the rest of the input features $\boldsymbol{x}$, and feed the resulting vector $[\boldsymbol{x}; f_1(\tau); \ldots; f_m(\tau)]$ to a sequence model (see Section 2 for references). This approach requires hand-crafting useful functions of time which may be difficult (or impossible) in several applications, and the hand-crafted functions may not be optimal for the task at hand. Instead of hand-crafting functions of time, we devise a representation of time which can be used to approximate any function through learnable parameters. Such a representation offers two advantages: 1- it obviates the need for hand-crafting functions of time and 2- it provides the grounds for learning suitable functions of time based on the data. As vector representations can be efficiently integrated with the current deep learning architectures, we employ a vector representation for time.

Our proposed representation leverages the Fourier sine series (Arfken & Weber, 1999) according to which any 1D function can be approximated in a given interval using a weighted sum of sinusoids with appropriate frequencies (and phase-shifts). We include $k$ sinusoids of the form $sin(\omega_i \tau + \varphi_i)$ in our vector representation where $\omega_i$ and $\varphi_i$ are learnable parameters[1]. That is, we concatenate the input features $\boldsymbol{x}$ with $k$ sinusoids and feed the concatenation $[\boldsymbol{x}; sin(\omega_1 \tau + \varphi_1); \ldots; sin(\omega_k \tau + \varphi_k)]$ into a sequence model. Different functions of time can be created using these sinusoids by taking a weighted sum of them with different weights. We allow the weights of the sequence model to combine the sinusoids and create functions of time suitable for the task. If we expand the output of the first layer of a sequence model (before applying an activation function), it has the form: $\boldsymbol{a}(\tau, k)[j] = \gamma_j + \sum_{i=1}^{k} \theta_{j,i} \sin(\omega_i \tau + \varphi_i)$, where $\theta_{j,i}$s are the first layer weights and $\gamma_j$ is the part of output which depends on the input features $\boldsymbol{x}$ (not on the temporal features). Each $\boldsymbol{a}(\tau, k)[j]$ operates on the input features $\boldsymbol{x}$ as well as a learned function $f_j(\tau) = \sum_{i=1}^{k} \theta_{j,i} \sin(\omega_i \tau + \varphi_i)$ of time, as opposed to a hand-crafted function[2]. Following Godfrey & Gashler (2018), to facilitate approximating functions with non-periodic patterns and help with generalization, we also include a linear projection of time in our vector representation. We name our vector representation of time *Time2Vec*. Time2Vec of $\tau$, denoted as $\mathbf{t2v}(\tau)$, is a vector of size $k + 1$ defined as follows:

$$\mathbf{t2v}(\tau)[i] = \begin{cases} \omega_i \tau + \varphi_i, & \text{if } i = 0. \\ \sin(\omega_i \tau + \varphi_i), & \text{if } 1 \leq i \leq k. \end{cases} \tag{1}$$

where $\mathbf{t2v}(\tau)[i]$ is the $i^{th}$ element of $\mathbf{t2v}(\tau)$ and $\omega_i$s and $\varphi_i$s are learnable parameters.

The use of sine functions is inspired in part by Vaswani et al. (2017)'s positional encoding. Consider a sequence of items (*e.g.*, a sequence of words) $\{I_1, I_2, \ldots, I_N\}$ and a vector representation $\boldsymbol{v}_{I_j} \in \mathbb{R}^d$ for the $j^{th}$ item $I_j$ in the sequence. Vaswani et al. (2017) added $\sin(j/10000^{k/d})$ to $\boldsymbol{v}_{I_j}[k]$ if $k$ is even and $\sin(j/10000^{k/d} + \pi/2)$ if $k$ is odd so that the resulting vector includes information about the position of the item in the sequence. These sine functions are called the positional encoding. Intuitively, positions can be considered as the times and the items can be considered as the events happening at that time. Thus, Time2Vec can be considered as representing continuous time, instead of discrete positions, using sine functions. The sine functions in Time2Vec also enable capturing periodic behaviors which is not a goal in positional encoding. We feed Time2Vec as an input to the model (or to some gate in the model) instead of adding it to other vector representations. Unlike positional encoding, we show in our experiments that learning the frequencies and phase-shifts of sine functions in Time2Vec result in better performance compared to fixing them.

---

[1] $k$ can be treated as a hyper-parameter.

[2] The function can model a real Fourier signal when the frequencies $\omega_i$ of the sine functions are integer multiples of a base (first harmonic) frequency. However, we show in Section 5.2 that learning the frequencies results in better generalization.

### 4.1 PROPERTIES OF TIME2VEC

We review some of the interesting and desired properties of Time2Vec.

**Periodicity:** In many scenarios, some events occur periodically. The amount of sales of a store, for instance, may be higher on weekends or holidays. Weather condition usually follows a periodic pattern over different seasons (Gashler & Ashmore, 2016). Some other events may be non-periodic but only happen after a point in time and/or become more probable as time proceeds. For instance, some diseases are more likely for older ages.

The period of $\sin(\omega_i \tau + \varphi_i)$ is $\frac{2\pi}{\omega_i}$, *i.e.* it has the same value for $\tau$ and $\tau + \frac{2\pi}{\omega_i}$. Therefore, the sine functions in Time2Vec help capture periodic behaviors without the need for feature engineering. For instance, a sine function $\sin(\omega \tau + \varphi)$ with $\omega = \frac{2\pi}{7}$ repeats every 7 days (assuming $\tau$ indicates days) and can be potentially used to model weekly patterns. Furthermore, unlike other basis functions which may show strange behaviors for extrapolation (see, *e.g.*, (Poole et al., 2014)), sine functions are expected to work well for extrapolating to future and out of sample data (Vaswani et al., 2017). The linear term represents the progression of time and can be used for capturing non-periodic patterns in the input that depend on time.

**Invariance to Time Rescaling:** Since time can be measured in different scales (*e.g.*, days, hours, seconds, etc.), another important property of a representation for time is invariance to time rescaling (see, *e.g.*, (Tallec & Ollivier, 2018)). A class $\mathcal{C}$ of models is invariant to time rescaling if for any model $\mathcal{M}_1 \in \mathcal{C}$ and any scalar $\alpha > 0$, there exists a model $\mathcal{M}_2 \in \mathcal{C}$ that behaves on $\alpha\tau$ ($\tau$ scaled by $\alpha$) in the same way $\mathcal{M}_1$ behaves on original $\tau$s. Proposition 1 establishes the invariance of Time2Vec to time rescaling. The proof is in Appendix D.

**Proposition 1.** *Time2Vec is invariant to time rescaling.*

**Simplicity:** A representation for time should be easily consumable by different models and architectures. A matrix representation, for instance, may be difficult to consume as it cannot be easily appended with the other inputs. By selecting a vector representation for time, we ensure easy integration with deep learning architectures.

## 5 EXPERIMENTS & RESULTS

We use the following datasets:

**1) Synthesized data:** We create a toy dataset to use for explanatory experiments. The inputs in this dataset are the integers between 1 and 365. Input integers that are multiples of 7 belong to class one and the other integers belong to class two. The first 75% is used for training and the last 25% for testing. This dataset is inspired by the periodic patterns (*e.g.*, weekly or monthly) that often exist in daily-collected data; the input integers can be considered as the days.

**2) Event-MNIST:** Sequential (event-based) MNIST is a common benchmark in sequence modeling literature (see, *e.g.*, (Bellec et al., 2018; Campos et al., 2018; Fatahi et al., 2016)). We create a sequential event-based version of MNIST by flattening the images and recording the position of the pixels whose intensities are larger than a threshold (0.9 in our experiment). Following this transformation, each image will be represented as an array of increasing numbers such as $[t_1, t_2, t_3, \ldots, t_m]$. We consider these values as the event times and use them to classify the images. As in other sequence modeling works, our aim in building this dataset is not to beat the state-of-the-art on the MNIST dataset; our aim is to provide a dataset where the only input is time and different representations for time can be compared when extraneous variables (confounders) are eliminated as much as possible.

**3) N_TIDIGITS18** (Anumula et al., 2018): The dataset includes audio spikes of the TIDIGITS spoken digit dataset (Leonard & Doddington, 1993) recorded by the binaural 64-channel silicon cochlea sensor. Each sample is a sequence of $(t, c)$ tuples where $t$ represents time and $c$ denotes the index of active frequency channel at time $t$. The labels are sequences of 1 to 7 connected digits with a vocabulary consisting of 11 digits (i.e. "zero" to "nine" plus "oh") and the goal is to classify the spoken digit based on the given sequence of active channels. We use the reduced version of the dataset where only the single digit samples are used for training and testing. The reduced dataset has a total of 2,464 training and 2,486 test samples.

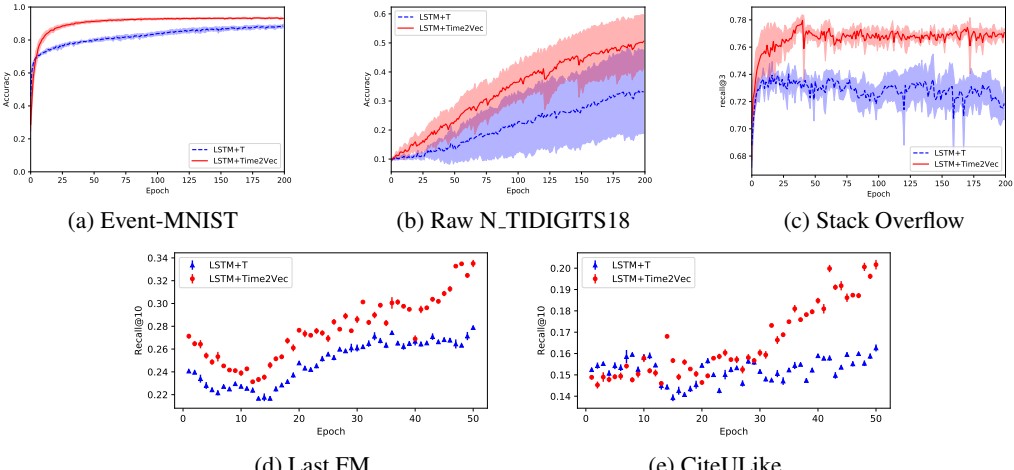

Figure 1: Comparing LSTM+T and LSTM+Time2Vec on several datasets.

**4) Stack Overflow (SOF):** This dataset contains sequences of badges obtained by stack overflow users and the timestamps at which the badges were obtained[3]. We used the subset released by Du et al. (2016) containing $\sim 6K$ users, 22 event types (badges), and $\sim 480K$ events. Given a sequence $[(b_1^u, t_1^u), (b_2^u, t_2^u), ..., (b_n^u, t_n^u)]$ for each user $u$ where $b_i^u$ is the badge id and $t_i^u$ is the timestamp when $u$ received this badge id, the task is to predict the badge the user will obtain at time $t_{k+1}^u$.

**5) Last.FM:** This dataset contains a history of listening habits for Last.FM users (Celma, 2010). We used the code released by Zhu et al. (2017) to pre-process the data. The dataset contains $\sim 1K$ users, 5000 event types (songs), and $\sim 819K$ events. The prediction problem is similar to the SOF dataset but with dynamic updating (see, (Zhu et al., 2017) for details).

**6) CiteULike:** This dataset contains data about what and when a user posted on citeulike website[4]. The original dataset has about 8000 samples. Similar to Last.FM, we used the pre-processing used by Zhu et al. (2017) to select $\sim 1.6K$ sequences with 5000 event types (papers) and $\sim 36K$ events. The task for this dataset is similar to that for Last.FM.

**Measures:** For classification tasks, we report *accuracy* corresponding to the percentage of correctly classified examples. For recommendation tasks, we report *Recall@q* and *MRR@q*. Following Zhu et al. (2017), to generate a recommendation list, we sample $k - 1$ random items and add the correct item to the sampled list resulting in a list of $k$ items. Then our model ranks these $k$ items. Looking only at the top ten recommendations, Recall@q corresponds to the percentage of recommendation lists where the correct item is in the top $q$; MRR@q (reported in Appendix B) corresponds to the mean of the inverses of the rankings of the correct items where the inverse rank is considered 0 if the item does not appear in top $q$ recommendations. For Last.FM and CiteULike, following Zhu et al. (2017) we report Recall@10 and MRR@10. For SOF, we report Recall@3 and MRR as there are only 22 event types and Recall@10 and MRR@10 are not informative enough. The detail of the implementations is presented in Appendix A.

### 5.1 ON THE EFFECTIVENESS OF TIME2VEC

Fig. 1 represents the obtained results of comparing *LSTM+Time2Vec* with *LSTM+T* on several datasets with different properties and statistics. On all datasets, replacing time with Time2Vec improves the performance in most cases and never deteriorates it; in many cases, LSTM+Time2Vec performs consistently better than LSTM+T. Anumula et al. (2018) mention that LSTM+T fails on N_TIDIGITS18 as the dataset contains very long sequences. By feeding better features to the LSTM rather than relying on the LSTM to extract them, Time2Vec helps better optimize the LSTM and

---

[3]https://archive.org/details/stackexchange
[4]http://www.citeulike.org/

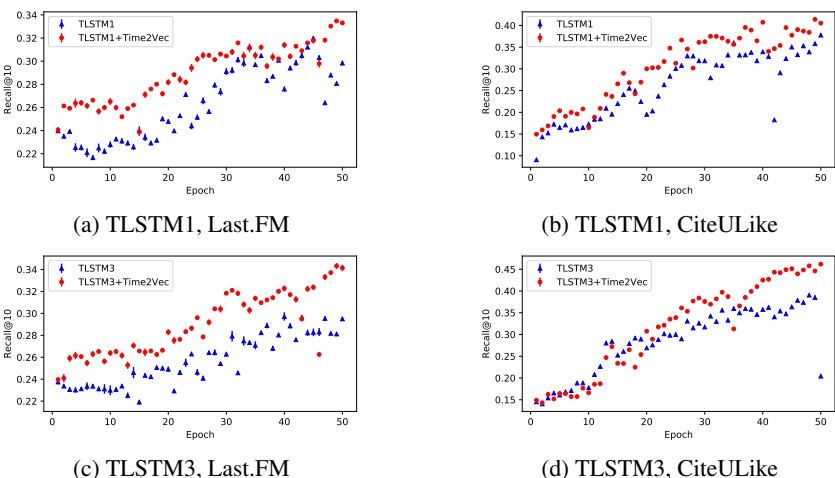

Figure 2: Comparing TLSTM1 and TLSTM3 on Last.FM and CiteULike in terms of Recall@10 with and without Time2Vec.

offers higher accuracy (and lower variance) compared to LSTM+T. Besides N_TIDIGITS18, SOF also contains somewhat long sequences and long time horizons. The results on these two datasets indicate that Time2Vec can be effective for datasets with long sequences and time horizons.

To verify if Time2Vec can be integrated with other architectures and improve their performance, we integrate it with TLSTM1 and TLSTM3, two recent and powerful models for handling asynchronous events. We replaced their notion $\tau$ of time with $\mathbf{t2v}(\tau)$ and replaced the vectors getting multiplied to $\tau$ with matrices accordingly. The updated formulations are presented in Appendix C. The obtained results in Fig. 2 for TLSTM1 and TLSTM3 on Last.FM and CiteULike demonstrates that replacing time with Time2Vec for both TLSTM1 and TLSTM3 improves the performance.

## 5.2 MODEL VARIANTS & ABLATION STUDY

**Other activation functions:** Inspired by Fourier sine series and by positional encoding, we used sine activations in Eq. 1. To evaluate how sine activations compare to other activation functions for our setting, we repeated the experiment on Event-MNIST in Section 5.1 when using non-periodic activations such as Sigmoid, Tanh, and rectified linear units (ReLU) (Nair & Hinton, 2010), and periodic activations such as *mod* and *triangle*. We fixed the length of the Time2Vec to $64 + 1$, *i.e.* 64 units with a non-linear transformation and 1 unit with a linear transformation. From the results shown in Fig. 5(a), it can be observed that the periodic activation functions (sine, mod, and triangle) outperform the non-periodic ones. Other than not being able to capture periodic behaviors, we believe one of the main reasons why these non-periodic activation functions do not perform well is because as time goes forward and becomes larger, Sigmoid and Tanh saturate and ReLU either goes to zero or explodes. Among periodic activation functions, sine outperforms the other two.

**Fixed frequencies and phase-shifts:** Vaswani et al. (2017) mention that learning sine frequencies and phase-shifts for their positional encoding gives the same performance as fixing frequencies to exponentially-decaying values and phase-shifts to 0 and $\frac{\pi}{2}$. This raises the question of whether learning the sine frequencies and phase-shifts of Time2Vec from data offer any advantage compared to fixing them. To answer this question, we compare three models on Event-MNIST when using Time2Vec of length $16 + 1$: 1- fixing $\mathbf{t2v}(\tau)[n]$ to $\sin\left(\frac{2\pi n}{16}\right)$ for $n \leq 16$, 2- fixing the frequencies and phase shifts according to Vaswani et al. (2017)'s positional encoding, and 3- learning the frequencies and phase-shifts from the data. Fig. 5(b) represents our obtained results. The obtained results in Fig. 5(b) show that learning the frequencies and phase-shifts rather than fixing them helps improve the performance of the model.

**Modeling Periodic Behaviours:** To measure how well Time2Vec performs in capturing periodic behaviours, we trained a model on our synthesized dataset where the input integer (day) is used as

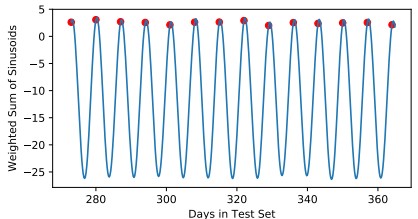
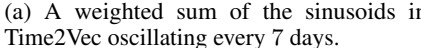
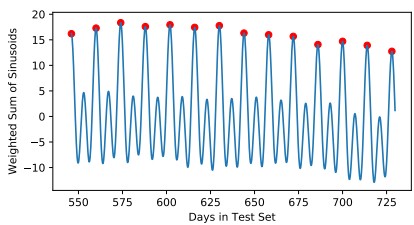

(a) A weighted sum of the sinusoids in Time2Vec oscillating every 7 days.

(b) A weighted sum of the sinusoids in Time2Vec oscillating every 14 days.

Figure 3: The models learned for our synthesized dataset before the final activation. The red dots represent the points to be classified as $1$.

the time for Time2Vec and a fully connected layer is used on top of the Time2Vec to predict the class. That is, the probability of one of the classes is a sigmoid of a weighted sum of the Time2Vec elements. Fig. 3 (a) shows a the learned function for the days in the test set where the weights, frequencies and phase-shifts are learned from the data. The red dots on the figure represent multiples of 7. It can be observed that Time2Vec successfully learns the correct period and oscillates every 7 days. The phase-shifts have been learned in a way that all multiples of 7 are placed on the positive peaks of the signal to facilitate separating them from the other days. Looking at the learned frequency and phase-shift for the sine functions across several runs, we observed that in many runs one of the main sine functions has a frequency around $0.898 \approx \frac{2\pi}{7}$ and a phase-shift around $1.56 \approx \frac{\pi}{2}$, thus learning to oscillate every 7 days and shifting by $\frac{\pi}{2}$ to make sure multiples of 7 end up at the peaks of the signal. Fig. 4 shows the initial and learned sine frequencies for one run. It can be viewed that at the beginning, the weights and frequencies are random numbers. But after training, only the desired frequency ($\frac{2\pi}{7}$) has a high weight (and the 0 frequency which gets subsumed into the bias). The model perfectly classifies the examples in the test set which represents the sine functions in Time2Vec can be used effectively for extrapolation and out of sample times assuming that the test set follows similar periodic patterns as the train set[5]. We added some noise to our labels by flipping $5\%$ of the labels selected at random and observed a similar performance in most runs.

To test invariance to time rescaling, we multiplied the inputs by 2 and observed that in many runs, the frequency of one of the main sine functions was around $0.448 \approx \frac{2\pi}{2*7}$ thus oscillating every 14 days. An example of a combination of signals learned to oscillate every 14 days is in Fig. 3 (b).

**The use of periodicity in sine functions:** It has been argued that when sine activations are used, only a monotonically increasing (or decreasing) part of it is used and the periodic part is ignored (Giambattista Parascandolo, 2017). When we use Time2Vec, however, the periodicity of the sine functions are also being used and seem to be key to the effectiveness of the Time2Vec representation. Fig. 5(c) shows some statistics on the frequencies learned for Event-MNIST where we count the number of learned frequencies that fall within intervals of lengths $0.1$ centered at $[0.05, 0.15, \ldots, 0.95]$ (all learned frequencies are between 0 and 1). The figure contains two peaks at $0.35$ and $0.85$. Since the input to the sine functions for this problem can have a maximum value of 784 (number

---

[5]Replacing sine with a non-periodic activation function resulted in always predicting the majority class.

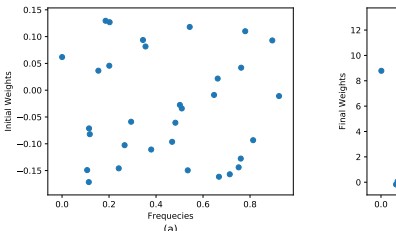
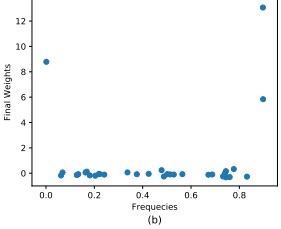

Figure 4: (a) Initial vs. (b) learned weights and frequencies for our synthesized dataset.

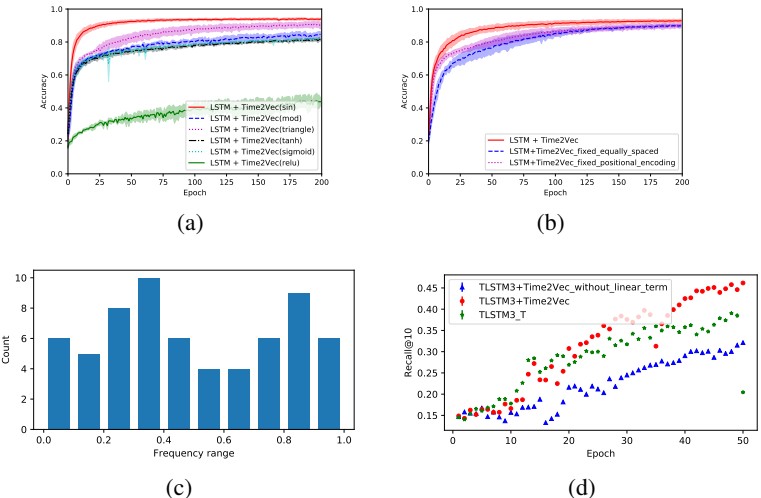

Figure 5: An ablation study of several components in Time2Vec. (a) Comparing different activation functions for Time2Vec on Event-MNIST. Sigmoid and Tanh almost overlap. (b) Comparing frequencies fixed to equally-spaced values, frequencies fixed according to positional encoding (Vaswani et al., 2017), and learned frequencies on Event-MNIST. (c) A histogram of the frequencies learned in Time2Vec for Event-MNIST. The x-axis represents frequency intervals and the y-axis represents the number of frequencies in that interval. (d) The performance of TLSTM3+Time2Vec on CiteULike in terms of Recall@10 with and without the linear term.

of pixels in an image), sine functions with frequencies around 0.35 and 0.85 finish (almost) 44 and 106 full periods. The smallest learned frequency is 0.029 which finishes (almost) 3.6 full periods. These values indicate that the model is indeed using the periodicity of the sine functions, not just a monotonically increasing (or decreasing) part of them.

**The Linear Term:** To see the effect of the linear term in Time2Vec, we repeated the experiment for Event-MNIST when the linear term is removed from Time2VecWe observed that the results were not affected substantially, thus showing that the linear term may not be helpful for Event-MNIST. This might be due to the simplicity of the Event-MNIST dataset. Then we conducted a similar experiment for TLSTM3 on CiteULike (which is a more challenging dataset) and obtained the results in Fig. 5(d). From these results, we can see that the linear term helps facilitate learning functions of time that can be effectively consumed by the model.

## 6 CONCLUSION & FUTURE WORK

In many tasks for synchronous and asynchronous event predictions, time is an important feature. Previous work has mainly resorted to applying hand-crafted functions to time and concatenating these functions with the rest of the input features. In this work, we presented an approach that automatically learns these functions from data. In particular, we developed Time2Vec, a vector representation for time, using sine and linear activations and showed the effectiveness of this representation across several datasets and several tasks. In the majority of our experiments, Time2Vec improved our results, while the remaining results were not hindered by its application. While sine functions have been argued to complicate the optimization (Lapedes & Farber, 1987; Giambattista Parascandolo, 2017), we did not experience such a complication except for the experiment in Subsection 5.2 on our synthesized dataset when using only a few sine functions. We hypothesize that the main reasons include combining sine functions with a powerful model (*e.g.*, LSTM) and using many sine functions which reduces the distance to the goal (see, *e.g.*, (Neyshabur et al., 2019)). We leave a deeper theoretical analysis of this hypothesis, development of better optimizers, and experimenting with other representations for time as future work.

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

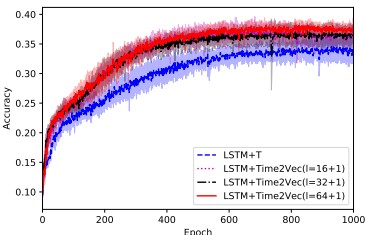

Figure 6: Comparing LSTM+T and LSTM+Time2Vec on Event-MNIST.

# A  IMPLEMENTATION DETAIL

For the experiments on Event-MNIST, N_TIDIGITS18 and SOF, we implemented[6] our model in PyTorch Paszke et al. (2017). We used Adam optimizer Kingma & Ba (2014) with a learning rate of $0.001$. For Event-MNIST and SOF, we fixed the hidden size of the LSTM to $128$. For N_TIDIGITS18, due to its smaller train set, we fixed the hidden size to $64$. We allowed each model $200$ epochs. We used a batch size of $512$ for Event-MNIST and $128$ for N_TIDIGITS18 and SOF. For the experiments on Last.FM and CiteULike, we used the code released by Zhu et al. (2017)[7] without any modifications, except replacing $\tau$ with $\mathbf{t2v}(\tau)$. The only other thing we changed in their code was to change the *SAMPLE_TIME* variable from 3 to 20. *SAMPLE_TIME* controls the number of times we do sampling to compute Recall@10 and MRR@10. We experienced a high variation when sampling only 3 times so we increased the number of times we sample to 20 to make the results more robust. For both Last.FM and CiteULike, Adagrad optimizer is used with a learning rate of $0.01$, vocabulary size is $5000$, and the maximum length of the sequence is $200$. For Last.FM, the hidden size of the LSTM is $128$ and for CiteULike, it is $256$. For all except the synthesized dataset, we shifted the event times such that the first event of each sequence starts at time $0$.

For the fairness of the experiments, we made sure the competing models for all our experiments have an (almost) equal number of parameters. For instance, since adding Time2Vec as an input to the LSTM increases the number of model parameters compared to just adding time as a feature, we reduced the hidden size of the LSTM for this model to ensure the number of model parameters stays (almost) the same. For the experiments involving Time2Vec, unless stated otherwise, we tried vectors with 16, 32 and 64 sine functions (and one linear term). We reported the vector length offering the best performance in the main text. The results for other vector lengths can be found in Appendix B. For the synthetic dataset, we use Adam optimizer with a learning rate of $0.001$ without any regularization. The length of the Time2Vec vector is 32.

---

[6]Code and datasets available at: `https://github.com/borealisai/Time2Vec`
[7]`https://github.com/DarryO/time_lstm`

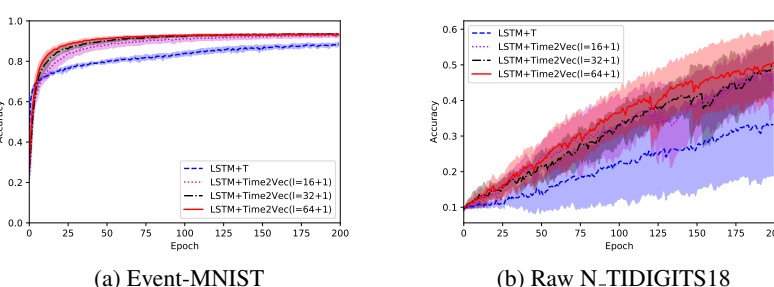

(a) Event-MNIST                         (b) Raw N_TIDIGITS18

Figure 7: Comparing LSTM+T and LSTM+Time2Vec on Event-MNIST and raw N_TIDIGITS18.

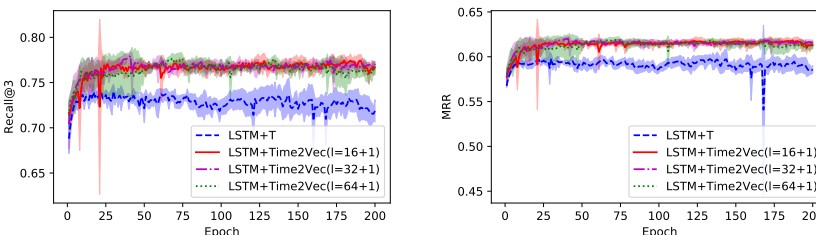

Figure 8: Comparing LSTM+T and LSTM+Time2Vec on SOF.

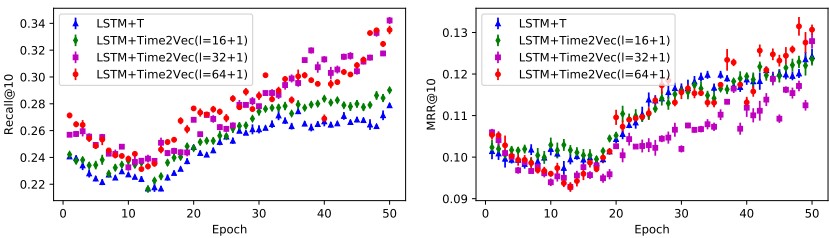

Figure 9: Comparing LSTM+T and LSTM+Time2Vec on Last.FM.

## B    MORE RESULTS

We ran experiments on other versions of the N_TIDIGITS18 dataset as well. Following Anumula et al. (2018), we converted the raw event data to *event-binned* features by virtue of aggregating active channels through a period of time in which a pre-defined number of events occur. The outcome of binning is thus consecutive frames each with multiple but a fixed number of active channels. In our experiments, we used event-binning with 100 events per frame. For this variant of the dataset, we compared LSTM+T and LSTM+Time2Vec similar to the experiments in Section 5.1. The obtained results were on-par. Then, similar to Event-MNIST, we only fed as input the times at which events occurred (*i.e.* we removed the channels from the input). We allowed the models 1000 epochs to make sure they converge. The obtained results are presented in Fig. 6. It can be viewed that Time2Vec provides an effective representation for time and LSTM+Time2Vec outperforms LSTM+T on this dataset.

In the main text, for the experiments involving Time2Vec, we tested Time2Vec vectors with 16, 32 and 64 sinusoids and reported the best one for the clarity of the diagrams. Here, we show the results for all frequencies. Figures 7, 8, 9, and 10 compare LSTM+T and LSTM+Time2Vec for our datasets. Figures 11, and 12 compare TLSTM1 with TLSTM1+Time2Vec on Last.FM and CiteULike. Figures 13, and 14 compare TLSTM3 with TLSTM1+Time2Vec on Last.FM and CiteULike. In most cases, Time2Vec with 64 sinusoids outperforms (or gives on-par results with) the cases with 32 or 16 sinusoids. An exception is TLSTM3 where 16 sinusoids works best. We believe that is because TLSTM3 has two time gates and adding, e.g., 64 temporal components (corresponding to the sinusoids) to each gate makes it overfit to the temporal signals.

## C    LSTM ARCHITECTURES

The original LSTM model can be neatly defined with the following equations:

$$\boldsymbol{i}_j = \sigma\left(\boldsymbol{W}_i\boldsymbol{x}_j + \boldsymbol{U}_i\boldsymbol{h}_{j-1} + \boldsymbol{b}_i\right) \tag{2}$$

$$\boldsymbol{f}_j = \sigma\left(\boldsymbol{W}_f\boldsymbol{x}_j + \boldsymbol{U}_f\boldsymbol{h}_{j-1} + \boldsymbol{b}_f\right) \tag{3}$$

$$\bar{\boldsymbol{c}}_j = Tanh\left(\boldsymbol{W}_c\boldsymbol{x}_j + \boldsymbol{U}_c\boldsymbol{h}_{j-1} + \boldsymbol{b}_c\right) \tag{4}$$

$$\boldsymbol{c}_j = \boldsymbol{f}_t \odot \boldsymbol{c}_{j-1} + \boldsymbol{i}_j \odot \overline{\boldsymbol{c_j}} \tag{5}$$

$$\boldsymbol{o}_j = \sigma\left(\boldsymbol{W}_o\boldsymbol{x}_j + \boldsymbol{U}_o\boldsymbol{h}_{j-1} + \boldsymbol{b}_o\right) \tag{6}$$

$$\boldsymbol{h}_j = \boldsymbol{o}_j \odot Tanh\left(\boldsymbol{c}_j\right) \tag{7}$$

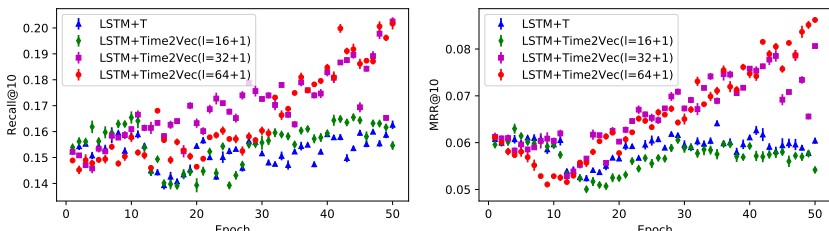

Figure 10: Comparing LSTM+T and LSTM+Time2Vec on CiteULike.

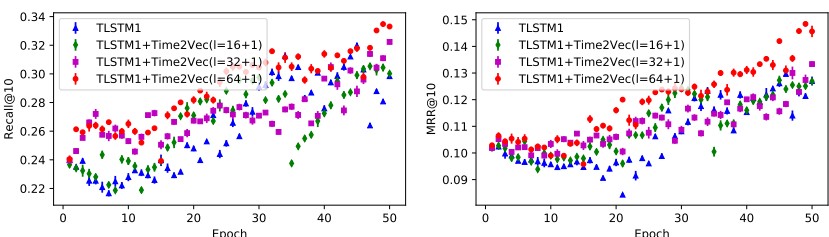

Figure 11: TLSTM1's performance on Last.FM with and without Time2Vec.

Here $\boldsymbol{i}_t$, $\boldsymbol{f}_t$, and $\boldsymbol{o}_t$ represent the input, forget and output gates respectively, while $\boldsymbol{c}_t$ is the memory cell and $\boldsymbol{h}_t$ is the hidden state. $\sigma$ and $Tanh$ represent the Sigmoid and hyperbolic tangent activation functions respectively. We refer to $\boldsymbol{x}_j$ as the $j^{th}$ event.

**Peepholes:** Gers & Schmidhuber (2000) introduced a variant of the LSTM architecture where the input, forget, and output gates peek into the memory cell. In this variant, $\boldsymbol{w}_{pi} \odot \boldsymbol{c}_{j-1}$, $\boldsymbol{w}_{pf} \odot \boldsymbol{c}_{j-1}$, and $\boldsymbol{w}_{po} \odot \boldsymbol{c}_j$ are added to the linear parts of Eq. (2), (3), and (6) respectively, where $\boldsymbol{w}_{pi}$, $\boldsymbol{w}_{pf}$, and $\boldsymbol{w}_{po}$ are learnable parameters.

**LSTM+T:** Let $\boldsymbol{\tau}_j$ represent the time features for the $j^{th}$ event in the input and let $\boldsymbol{x}'_j = [\boldsymbol{x}_j; \boldsymbol{\tau}_j]$. Then LSTM+T uses the exact same equations as the standard LSTM (denoted above) except that $\boldsymbol{x}_j$ is replaced with $\boldsymbol{x}'_j$.

**TimeLSTM:** We explain TLSTM1 and TLSTM3 which have been used in our experiments. For clarity of writing, we do not include the peephole terms in the equations but they are used in the experiments. In TLSTM1, a new time gate is introduced as in Eq. equation 8 and Eq. equation 5 and equation 6 are updated to Eq. equation 9 and equation 10 respectively:

$$\boldsymbol{t}_j = \sigma \left( \boldsymbol{W}_t \boldsymbol{x}_j + \sigma \left( \boldsymbol{u}_t \tau_j \right) + \boldsymbol{b}_t \right) \tag{8}$$

$$\boldsymbol{c}_j = \boldsymbol{f}_j \odot \boldsymbol{c}_{j-1} + \boldsymbol{i}_j \odot \boldsymbol{t}_j \odot \overline{\boldsymbol{c}_j} \tag{9}$$

$$\boldsymbol{o}_j = \sigma \left( \boldsymbol{W}_o \boldsymbol{x}_j + \boldsymbol{v}_t \tau_j + \boldsymbol{U}_o \boldsymbol{h}_{j-1} + \boldsymbol{b}_o \right) \tag{10}$$

$\boldsymbol{t}_j$ controls the influence of the current input on the prediction and makes the required information from timing history get stored on the cell state. TLSTM3 uses two time gates:

$$\boldsymbol{t1}_j = \sigma \left( \boldsymbol{W}_{t1} \boldsymbol{x}_j + \sigma \left( \boldsymbol{u}_{t1} \tau_j \right) + \boldsymbol{b}_{t1} \right) \tag{11}$$

$$\boldsymbol{t2}_j = \sigma \left( \boldsymbol{W}_{t2} \boldsymbol{x}_j + \sigma \left( \boldsymbol{u}_{t2} \tau_j \right) + \boldsymbol{b}_{t2} \right) \tag{12}$$

where the elements of $\boldsymbol{W}_{t1}$ are constrained to be non-positive. $\boldsymbol{t1}$ is used for controlling the influence of the last consumed item and $\boldsymbol{t2}$ stores the $\tau$s thus enabling modeling long range dependencies. TLSTM3 couples the input and forget gates following Greff et al. (2017) along with the $\boldsymbol{t1}$ and $\boldsymbol{t2}$ gates and replaces Eq. (5) to (7) with the following:

$$\widetilde{\boldsymbol{c}}_j = \left( 1 - \boldsymbol{i}_j \odot \boldsymbol{t1}_j \right) \odot \boldsymbol{c}_{j-1} + \boldsymbol{i}_j \odot \boldsymbol{t1}_j \odot \bar{\boldsymbol{c}}_j \tag{13}$$

$$\boldsymbol{c}_j = \left( 1 - \boldsymbol{i}_j \right) \odot \boldsymbol{c}_{j-1} + \boldsymbol{i}_j \odot \boldsymbol{t2}_j \odot \bar{\boldsymbol{c}}_j \tag{14}$$

$$\boldsymbol{o}_j = \sigma \left( \boldsymbol{W}_o \boldsymbol{x}_j + \boldsymbol{v}_t \tau_j + \boldsymbol{U}_o \boldsymbol{h}_{j-1} + \boldsymbol{b}_o \right) \tag{15}$$

$$\boldsymbol{h}_j = \boldsymbol{o}_j \odot Tanh \left( \widetilde{\boldsymbol{c}}_j \right) \tag{16}$$

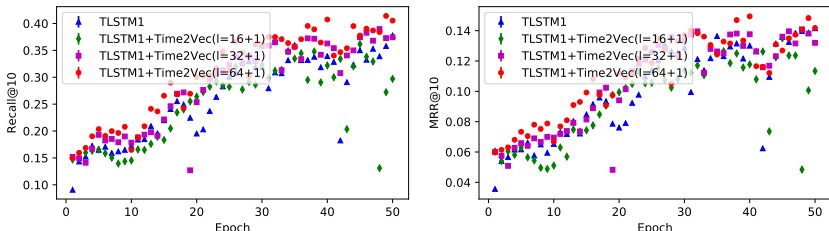

Figure 12: TLSTM1's performance on CiteULike with and without Time2Vec.

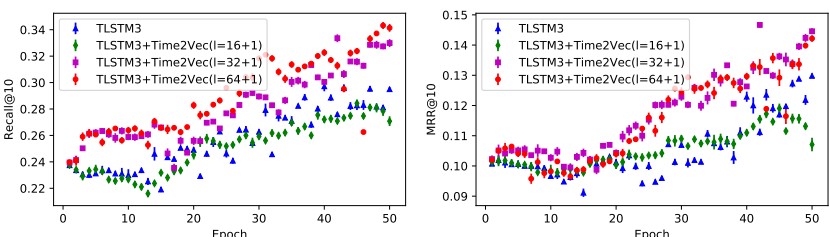

Figure 13: TLSTM3's performance on Last.FM with and without Time2Vec.

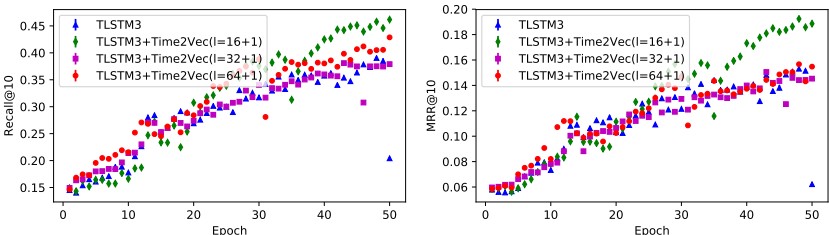

Figure 14: TLSTM3's performance on CiteULike with and without Time2Vec.

Zhu et al. (2017) use $\tau_j = \Delta t_j$ in their experiments, where $\Delta t_j$ is the duration between the current and the last event.

**TimeLSTM+Time2Vec:** To replace time in TLSTM1 with Time2Vec, we modify Eq. (8) and (10) as follows:

$$\boldsymbol{t}_j = \sigma\left(\boldsymbol{W}_t\boldsymbol{x}_j + \sigma\left(\boldsymbol{U}_t\mathbf{t2v}(\tau)\right) + \boldsymbol{b}_t\right) \tag{17}$$

$$\boldsymbol{o}_j = \sigma(\boldsymbol{W}_o\boldsymbol{x}_j + \boldsymbol{V}_t\mathbf{t2v}(\tau) + \boldsymbol{U}_o\boldsymbol{h}_{j-1} + \boldsymbol{b}_o) \tag{18}$$

i.e., $\tau$ is replaced with $\mathbf{t2v}(\tau)$, $\boldsymbol{u}_t$ is replaced with $\boldsymbol{U}_t$, and $\boldsymbol{v}_t$ is replaced with $\boldsymbol{V}_t$. Similarly, for TLSTM3 we modify Eq. (11), (12) and (15) as follows:

$$\boldsymbol{t1}_j = \sigma\left(\boldsymbol{W}_{t1}\boldsymbol{x}_j + \sigma\left(\boldsymbol{U}_{t1}\mathbf{t2v}(\tau)\right) + \boldsymbol{b}_{t1}\right) \tag{19}$$

$$\boldsymbol{t2}_j = \sigma\left(\boldsymbol{W}_{t2}\boldsymbol{x}_j + \sigma\left(\boldsymbol{U}_{t2}\mathbf{t2v}(\tau)\right) + \boldsymbol{b}_{t2}\right) \tag{20}$$

$$\boldsymbol{o}_j = \sigma(\boldsymbol{W}_o\boldsymbol{x}_j + \boldsymbol{V}_t\mathbf{t2v}(\tau) + \boldsymbol{U}_o\boldsymbol{h}_{j-1} + \boldsymbol{b}_o) \tag{21}$$

## D  PROOFS

**Proposition 1.** *Time2Vec is invariant to time rescaling.*

*Proof.* Consider the following Time2Vec representation $\mathcal{M}_1$:

$$\mathbf{t2v}(\tau)[i] = \begin{cases} \omega_i\tau + \varphi_i, & \text{if } i = 0. \\ \sin\left(\omega_i\tau + \varphi_i\right), & \text{if } 1 \le i \le k. \end{cases} \tag{22}$$

Replacing $\tau$ with $\alpha \cdot \tau$ (for $\alpha > 0$), the Time2Vec representation updates as follows:

$$\mathbf{t2v}(\alpha \cdot \tau)[i] = \begin{cases} \omega_i(\alpha \cdot \tau) + \varphi_i, & \text{if } i = 0. \\ \sin\left(\omega_i(\alpha \cdot \tau) + \varphi_i\right), & \text{if } 1 \leq i \leq k. \end{cases} \tag{23}$$

Consider another Time2Vec representation $\mathcal{M}_2$ with frequencies $\omega_i' = \frac{\omega_i}{\alpha}$. Then $\mathcal{M}_2$ behaves in the same way as $\mathcal{M}_1$. This proves that Time2Vec is invariant to time rescaling. $\square$

