# OpenReview forum: "Time2Vec: Learning a Vector Representation of Time"
_ICLR.cc/2020/Conference — Reject_

### Official Review · AnonReviewer1 · 2019-10-23
**Official Blind Review #1**

**Rating:** 6

**Review:**

# Summary
This paper proposes a simple representation of time (Time2Vec) for modelling sequential data. The idea is to apply multiple sine functions to the time with trainable period and offset and concatenate them together, which is similar to positional encoding [Vaswani et al.] except that the periods and offsets are learned. The results on several sequential modelling datasets show that Time2Vec performs better than naive representations and alternative baselines.

# Originality
- Although the proposed representation looks simple and similar to positional encoding [Vaswani et al.], the idea of parameterizing sine functions is novel and interesting.

# Quality
- The proposed idea is very simple but seems very effective in practice as shown by the empirical results.
- The paper also provides in-depth analysis and ablation studies showing that each of the proposed component is helpful.
- Although the results presented in this paper look very promising, it would be much stronger if the paper presented results on other sequential modelling tasks such as machine translation and language modelling that the research community cares about much more. For example, it would be great if the paper showed that replacing the fixed positional encoding with Time2Vec improves the performance on a machine translation dataset.
- It would be good to show the effect of the number of sine units (k) in Time2Vec.
- (Minor) Clockwork RNN [Koutn´ık et al.] introduces a nice toy periodic sequential prediction problem, where the model has to recover a mixture of sine/cosine function. It looks like a natural task to show in this paper as well (which can potentially replace the synthesized data experiment).

# Clarity
- The paper is overall well-written.
- Figure 4 is not mentioned in the main text.
- Figure 5 is mentioned earlier than Figure 3. It would be good to swap them.

# Significance
- This paper proposes a simple but effective idea that can be potentially widely used by the research community. The paper would be stronger if it included more results on high-impact sequential modelling tasks and datasets such as machine translation.

**Experience Assessment:**

I have read many papers in this area.

**Review Assessment: Checking Correctness Of Derivations And Theory:**

I assessed the sensibility of the derivations and theory.

**Review Assessment: Checking Correctness Of Experiments:**

I assessed the sensibility of the experiments.

**Review Assessment: Thoroughness In Paper Reading:**

I read the paper at least twice and used my best judgement in assessing the paper.

---

> ### Author Response · Authors · 2019-11-13
> **Response**
>
> We thank the reviewer for valuable feedback.
>
> Positional encoding, the analog of Time2Vec for positions, has been extensively used and studied for neural machine translation and its merit has been established in that community. Our work extends the merit of a representation similar to positional encoding (but with a different motivation) to the time-series community.
>
> The effect of the number of sine functions can be viewed in the supplementary. We tried 16, 32, and 64 sine functions for our experiments. Except in one case, Time2Vec with 64 frequencies outperformed Time2Vec with 16 or 32 frequencies in other cases. This can be due to reducing the distance to the goal (see the conclusion section) or due to enabling better approximations (according to the Fourier sine series) when creating functions of time.
>
> Thanks for introducing the experiment in Clockwork RNN paper to us. That indeed seems like an interesting experiment for us and we will acknowledge that in our paper. The reason for using a hand-crafted dataset was because we could control the underlying dynamics and verify if the model can learn the correct dynamics. Alternatively, we could do the experiment suggested by the reviewer (i.e. recovering a mixture of sine functions).
>
> Figure 4 is mentioned in the main text at the bottom of the first paragraph on page 7 (“…Fig. 4 shows the initial and learned sine frequencies…”).

---

### Official Review · AnonReviewer3 · 2019-10-23
**Official Blind Review #3**

**Rating:** 1

**Review:**

## Summary

The authors propose a method for encoding time features using a sine function with learned phase and frequency. They apply this method to several synthetic and real-world datasets.

Temporal and positional encoding is important to many applications, including NLP, sound understanding and time series modeling, so the topic is certainly of interest. However, the method they propose offers very little that is new when compared to e.g. Vaswani (https://arxiv.org/pdf/1706.03762.pdf, section 3.5) (the authors acknowledge this work several times). In addition, the authors compare to a baseline that seems to consist of passing time as a float. This seems like a very weak baseline---there are any number of other reasonable ways to encode this.

Due to the incremental nature of the improvement and the weak baseline, I don't think this paper should be accepted to ICLR.


## Specific Comments

1. In Section 2, I find the sentence "We follow a similar intuition but instead of decomposing a 1D signal of time into its components, we transform the time itself and feed its transformation into the model that is to consume the time information" really unclear. Could you rephrase it?

2. Often, positional encodings are used to encode ordering for a model architecture that is not inherently sequential. This is the case for the positional encodings in the transformer model. Did you try these encodings with non-recurrent architectures?

3. In Section 5.2, did you mean 'fixing t2v(\tau)[n] = sin(2\pi n \tau / 16)'? i.e. I think it's missing a 'tau'

4. In Section 5.2 "Fixed frequencies and phase shifts" you compare Time2Vec to a fixed set of frequencies. Since positional encoding with Fourier transforms is well known, this seems like the relevant benchmark but it receives only a brief treatment. The authors compare these methods only on Event-MNIST and only for 16 frequencies. I would like to see this comparison expanded.

5. Could you clarify exactly how time is encoded for LSTM + T? Are you, in fact, just passing a float value? How is this encoded for each data set? For example, the "times" for Event-MNIST is always [0, 783] while the SOF data has timestamps. What is the encoding scheme for each?

**Experience Assessment:**

I have read many papers in this area.

**Review Assessment: Checking Correctness Of Derivations And Theory:**

N/A

**Review Assessment: Checking Correctness Of Experiments:**

I assessed the sensibility of the experiments.

**Review Assessment: Thoroughness In Paper Reading:**

I read the paper at least twice and used my best judgement in assessing the paper.

---

> ### Author Response · Authors · 2019-11-13
> **Response**
>
> We thank the reviewer for valuable feedback.
>
> “the method they propose offers very little that is new when compared to e.g. Vaswani”
> While the final representation of Time2Vec resembles that of positional encoding, the motivation behind Time2Vec is completely different than that of positional encoding. The new things offered by Time2Vec compared to positional encoding and other previous work include:
>
>  - Instead of using time as a scalar feature similar to other features (which as the reviewer also pointed out is a naive way of handling time), we identify the characteristics that differentiate “time” from other features and propose a representation that enables exploiting those characteristics. Note that using time as a scalar feature similar to other features is currently the prevalent choice (see the references in the last paragraph of section 2).
>
>  - Obviating the need for hand-crafting functions of time by instead enabling these functions to be learned from data, and backing up the representation theoretically as, according to Fourier sine series, it can approximate any function in a given interval (see the last paragraph of our response to reviewer 5).
>
>  - Providing a comprehensive set of experiments showing the merit of Time2Vec for time-series prediction problems where time is an important feature. This includes, among other things, results for modeling periodic behaviors of signals which is not a goal in positional encoding.
>
> Although our representation resembles positional encoding on the surface, it has not been clear in the time-series community if/how/why positional encoding can be used to replace hand-crafted functions of time, and there has been no empirical evidence to show its merit.
>
> “Weak baseline”
> Our goal is to propose a representation of time that can be used instead of merely a float notion of time (as opposed to beating the state-of-the-art on a particular dataset). Therefore, all our comparisons are head-to-head comparisons between a model with and without Time2Vec. This includes LSTM+T vs LSTM+Time2Vec, TimeLSTM1 vs TimeLSTM1 + Time2Vec, and TimeLSTM3 vs TimeLSTM3 + Time2Vec. It would not be sensible to, e.g., compare LSTM+Time2Vec to TimeLSTM3 (or some other model) because the results of such an experiment do not provide evidence towards Time2Vec being useful or useless.
>
> Specific comments:
> We will clarify the sentence in section 2. Except for the hand-crafted experiment, we did not use Time2Vec in non-recurrent architectures. In Section 5.2, a \tau is indeed missing; we’ll fix this. If the paper gets accepted, we will expand the experiments with fixed frequencies in the final version. In both LSTM+T and LSTM+Time2Vec, for Event-MNIST \tau is a feature between [0, 783], for NT-DIGITS and SOF \tau corresponds to the Unix timestamps when events occurred, and in LastFM and CiteULike \tau corresponds to the delta between the current and previous event.

---

> > ### Comment · AnonReviewer3 · 2019-11-15
> > **Response**
> >
> > The main concern I have with this paper reamains---it adds very little to existing methods for positional/temporal encoding. As I (and the authors) mentioned, Vaswani et al. used a sinusoids to encode position and even suggested learning the frequencies. I don't think this paper adds enough to this idea to warrant accepting it to ICLR.
> >
> >
> > ## "While the final representation of Time2Vec resembles that of positional encoding, the motivation behind Time2Vec is completely different than that of positional encoding."  ##
> >
> > I disagree with this. Vaswani et al. describe general 'Positional Encoding.' Time seems like a very natural type of position to encode.
> >
> >
> > ## "Note that using time as a scalar feature similar to other features is currently the prevalent choice (see the references in the last paragraph of section 2)."
> >
> > I don't think passing scalars such as Unix time stamps is the prevalent choice, particularly for a data set like 'Stack Overflow.' For irregularly spaced sequences I think using (a function of) deltas between the sequences is much more common, and the authors mention this approach in section 2.
> >
> >
> > ## Baselines
> >
> > The novelty of this paper is using *learned* frequencies and phases rather than fixed ones (although this idea is mentioned by Vaswani...). I think the relevant comparison is between an architecture equipped with time2vec and the same architecture equipped with fixed frequency sinusoids.
> >
> > From the authors' response, it seems that time was encoded using 'deltas' for some data sets (LastFM and CiteULike) and absolute time for others (MNIST, NT-DIGITS, SOF). As a result, the models they compare to (e.g. LSTM + T) are not consistent across different data sets.

---

### Official Review · AnonReviewer5 · 2019-11-02
**Official Blind Review #5**

**Rating:** 3

**Review:**

This paper introduces a particular learnable vector representation of time which is applicable across problems without the use of a hand-crafted time representation. Their representation makes use of a feed-forward layer with sine activations which operates on time data. As it is a vector representation, it combines well with other deep neural network methods. They motivate their problem well, explaining why time data is important to a variety of problems and situate their solution as an orthogonal approach to many current solutions in the literature. They make reference to fourier analysis as motivation for their representation. Finally, they provide experimental results to support their claims using fabricated and real-world time series datasets, as well as ablation studies to support their design decisions.

While I think this work has the potential to be a significant contribution, I rate this a weak reject because the theoretical motivation and analysis of the experimental results are lacking the depth of evidence I would expect for an ICLR paper. If you provide a deeper discussion of the provable claims about the power of your model via Fourier analysis and provide a table of test accuracy/recall@K with/without your representation for more than one other state of the art algorithm for these datasets, I would be convinced to strong accept.

Specific comments:

* p.3 third paragraph: you repeat yourself in math notation a few times here. Repeated equations usually indicate that there is something new happening, but all of these are just restatements of your theta sin(omega tau + phi) term. I would introduce the notation for t2v(tau) upfront and use that to define a(tau, k)[j] and f_j
* p.3 A clearer explanation of the theory here would help, as I think Fourier's theorem nicely supports your claims.
* p.4 first paragraph you claim that this method responds well to data which exhibits seasonality, but none of your datasets deal with data that would exhibit seasonality. There are plenty of simple real-world datasets available which show multi-scale periodic phenomena (activity or location data, weather data, travel data, etc.). In fact, segmentation and recognition of wearable device activity would be a great application for this method.
* p.4 third paragraph: Your claim of invariance to time rescaling is technically correct, but I am not convinced that a model can learn the correct omega values for an arbitrary rescaling (e.g. if the period is smaller than the time unit). You show that this works for a rescaling from 2pi/7 to 2pi/14, but it would be nice if there was experimental confirmation of this property with frequency > 1.
* p.6 Showing accuracy/recall across training epochs is not sufficient evidence to show that this is a useful representation. There should be some kind of comparison with test set results from other state-of-the-art work on these datasets. If adding your representation to the SOTA model improved test set performance (or at least sped up training without hurting test set performance), then that would be better evidence. If LSTM+T is the SOTA, say so and restate the author's test performance compared to yours. If this is what these graphs show, consider using a different visualization to make it clearer that you're improving the final performance, not just the training process.
* p.8 I think sine functions make optimization harder because they make the gradient function periodic with respect to the weights, creating infinitely many local extrema. Historically this may have been an issue, but deep neural networks have so many local minima it might not matter. Still, it would be good to show that trained performance doesn't depend on the initialization values more than a standard LSTM+T model.
* You have an interesting corner case where your neural network parameters are interpretable: you can interpret the omega values from your model as frequencies and investigate their values to see which kinds of periodicity your model uses. You do something like this on p.7, but it would be neat to see a histogram like the one you have for EventMNIST for one of the real-world datasets to see if it learns the domain-relevant time knowledge you claim that it should learn.

**Experience Assessment:**

I have read many papers in this area.

**Review Assessment: Checking Correctness Of Derivations And Theory:**

I assessed the sensibility of the derivations and theory.

**Review Assessment: Checking Correctness Of Experiments:**

I assessed the sensibility of the experiments.

**Review Assessment: Thoroughness In Paper Reading:**

I read the paper thoroughly.

---

> ### Author Response · Authors · 2019-11-14
> **Response**
>
> We would like to thank the reviewer for constructive feedback.
>
> Results: We would like to clarify that all the results reported in the paper are on test sets (this includes Figures 1, 2, 3, and 5 as well as those in the supplementary). We decided to report the test set performance for all epochs instead of just the last epoch to show that: 1- in many cases, LSTM+Time2Vec consistently outperforms LSTM+T, 2- replacing the notion of time with Time2Vec does not deteriorate the performance, and 3- adding Time2Vec makes the model reach its best performance faster. Sorry about the confusion, we will clarify this in the paper.
> “If adding your representation to the SOTA model improved test set performance (or at least sped up training without hurting test set performance), then that would be better evidence.” ->  This is indeed what we did. We showed that adding Time2Vec to LSTM+T (the model used in several recent works - see the last paragraph of related works section) and to two variants of TimeLSTM (a recent architecture with remarkable results on asynchronous sequential datasets) improves test set performance.
> “test accuracy/recall@K with/without your representation for more than one other state of the art algorithm for these datasets”: Upon the reviewer’s request, we are looking to extend one more architecture with Time2Vec. If we managed to obtain results until the end of the rebuttal period, we will post them here.
> Dataset that exhibits seasonality: The hand-crafted dataset has been created to serve that purpose (we could change the frequency from weekly to monthly or quarterly). The reason for using a hand-crafted dataset was because we could control the underlying dynamics and verify if the model can learn the correct dynamics.
>
> Optimization of sine functions: The results we have reported in the paper demonstrate mean and standard deviation across multiple runs. In each run, we initialize the parameters randomly. The standard deviations provide evidence that the performance of LSTM+Time2Vec doesn't depend on the initialization values more than a standard LSTM+T model. Moreover, from Fig 1(b) and 1(c), it can be observed that the standard deviation of LSTM+Time2Vec is even smaller than that of LSTM+T.
>
> Theory: According to Fourier sine-cosine series, any real-valued function f(t) that is integrable on an interval of length P can be approximated as f(t) = a_0 + sum_{n=1}^{N/2} (a_n cos(2nt\pi/P) + b_n sin(2nt\pi/P)) by choosing appropriate weights a_n and b_n. Since cos(x)=sin(x+\pi/2), the cos functions can be replaced with sine functions so f(t) can be approximated with N sine functions.
> By concatenating Time2Vec to the input, as explained in the second paragraph of Section 4, we allow the sequence model to learn a function (or multiple functions) of time based on the data by taking a weighted sum (the weights correspond to a_n and b_n in the formula above) of the sinusoids. Learning a function of time from data rather than fixing it to a hand-crafted function can potentially lead to better generalization.
> We will state the theory behind Time2Vec more explicitly.

---

### Official Review · AnonReviewer4 · 2019-11-02
**Official Blind Review #4**

**Rating:** 8

**Review:**

The authors proposed a new embedding for time - Time2Vec. Unlike previous research that is either proposing a new architecture or proposing expensive handcrafted features, this work proposes a model-agnostic learnable time embedding.

I would like to recommend an accept based on the following reasons:
* Modeling time is crucial for quite a few machine learning tasks. With the two most desired properties, learnable and model-agnostic, this time embedding will be very useful in various applications.
* The authors are good at story-telling and this makes the paper very readable and approachable. This increases the chance of the contribution made in this paper to be applied in real-world applications.
* This work did clear and detailed analysis on both the empirical results and the probing experiments.




**Experience Assessment:**

I have read many papers in this area.

**Review Assessment: Checking Correctness Of Derivations And Theory:**

I assessed the sensibility of the derivations and theory.

**Review Assessment: Checking Correctness Of Experiments:**

I carefully checked the experiments.

**Review Assessment: Thoroughness In Paper Reading:**

I read the paper thoroughly.

---

> ### Author Response · Authors · 2019-11-13
> **Response**
>
> We thank the reviewer for their feedback.

---

### Public Comment · ~Eelco_Hoogendoorn1 · 2020-04-12
**A word of encouragement**

Id like to add a word of encouragement to the authors.

Even if we should assume there isnt a single 'novel' idea in this paper (and I am not saying there isnt), given how highly under-studied the subject of this paper is, any general reflection on the question of how to best encode time/position in a modern machine learning context, is a much more valuable contribution, than the vast majority of published pages, of authors contorting themselves to prove the 'novelty' of their architectural gizmos.

Yes, I can think of a million more experiments and theoretical questions to be pursued along the lines of this paper as well. But until people can point at other papers that already covered the ground this paper does, that seems like an exceedingly poor reason for rejection, given the complete lack of attention given in the literature, to this subject of tremendous practical importance.

Yes, maybe this is just an investigation of how a slight variant of positional encoding popularised by attention networks performs in other contexts. But if there was a single paper id read in the next year, that would probably be it.

---

### Decision · Program_Chairs · 2019-12-19

**Decision:**

Reject

**Comment:**

This paper investigates and evaluates learning high-dimensional embeddings of time, which is useful for a variety of applications. This paper received 4 reviews (due to a missing review, we requested several emergency reviews). R1 recommends Weak Accept, calling the method simple but saying it could be of wide interest and utility in practice. R3 recommends Reject, identifying concerns about the significance of the contribution, caused by the simplicity of the approach, the connection to existing work, and missing comparisons to baselines. In a short review, R4 recommends Accept with several positive comments. In a long, thoughtful review, R5 recommends Weak Reject, due to concerns and questions about the theoretical motivation and depth of experiments. The authors have submitted detailed responses that have addressed many of the questions of the reviewers; however, R3 feels the response does not address their concerns, and R5 is closer to accepting but still feels additional improvements in presentation and experimentation are needed.

Given the split decision, the AC also read the paper. The AC agrees with R1 and R4 that this is an interesting problem and the approach here may be useful in practice, but shares concerns with R3 and R5 about the depth of contribution with respect to existing work, and need for additional experimental validation against stronger baselines.